# CAN NEURAL NETWORKS UNDERSTAND LOGICAL ENTAILMENT?

**Richard Evans**[*]      **David Saxton**[*]      **David Amos**      **Pushmeet Kohli**

**Edward Grefenstette**[*]
DeepMind
{richardevans,saxton,davidamos,pushmeet,etg}@google.com

## ABSTRACT

We introduce a new dataset of logical entailments for the purpose of measuring models' ability to capture and exploit the structure of logical expressions against an entailment prediction task. We use this task to compare a series of architectures which are ubiquitous in the sequence-processing literature, in addition to a new model class—PossibleWorldNets—which computes entailment as a "convolution over possible worlds". Results show that convolutional networks present the wrong inductive bias for this class of problems relative to LSTM RNNs, tree-structured neural networks outperform LSTM RNNs due to their enhanced ability to exploit the syntax of logic, and PossibleWorldNets outperform all benchmarks.

## 1 INTRODUCTION

This paper seeks to answer two questions: "Can neural networks understand logical formulae well enough to detect entailment?", and, more generally, "Which architectures are best at inferring, encoding, and relating features in a purely structural sequence-based problem?". In answering these questions, we aim to better understand the inductive biases of popular architectures with regard to structure and abstraction in sequence data. Such understanding would help pave the road to agents and classifiers that reason structurally, in addition to reasoning on the basis of essentially semantic representations. In this paper, we provide a testbed for evaluating some aspects of neural networks' ability to reason structurally and abstractly. We use it to compare a variety of popular network architectures and a new model we introduce, called PossibleWorldNet.

Neural network architectures lie at the heart of a variety of applications. They are practically ubiquitous across vision tasks (LeCun et al., 1995; Krizhevsky et al., 2012; Simonyan & Zisserman, 2014) and natural language understanding, from machine translation (Kalchbrenner & Blunsom, 2013; Sutskever et al., 2014; Bahdanau et al., 2014) to textual entailment (Bowman et al., 2015; Rocktäschel et al., 2015) via sentiment analysis (Socher et al., 2013; Kalchbrenner et al., 2014) and reading comprehension (Hermann et al., 2015; Hill et al., 2015; Rajpurkar et al., 2016). They have been used to synthesise programs (Ling et al., 2016; Parisotto et al., 2016; Devlin et al., 2017) or internalise algorithms (Graves et al., 2016; Grefenstette et al., 2015; Joulin & Mikolov, 2015; Kaiser & Sutskever, 2015; Reed & De Freitas, 2015). They form the basis of reinforcement learning agents capable of playing video games (Mnih et al., 2015), difficult perfect information games (Silver et al., 2016; Tian & Zhu, 2015), and navigating complex environments from raw pixels (Mirowski et al., 2016). An important question in this context is to find the inductive and generalisation properties of different neural architectures, particularly towards the ability to capture structure present in the input, an ability that might be important for many language and reasoning tasks. However, there is little work on studying these inductive biases in isolation by running these models on tasks that are primarily or purely *about* sequence structure, which we intend to address.

The paper's contribution is three-fold. First, we introduce a new dataset for training and evaluating models. Second, we provide a thorough evaluation of the existing neural models on this dataset. Third, inspired by the semantic (model-theoretic) definition of entailment, we propose a variant of

---

[*]Equal contribution.

the TreeNet that evaluates the formulas in multiple different "possible worlds", and which significantly outperforms the benchmarks. The structure of this paper is as follows. In Section 2, we introduce the new dataset and describe a generic data generation process for entailment datasets, which offers certain guarantees against the presence of superficial exploitable biases. In Section 3, we describe a series of baseline models used to validate the dataset, benchmarks from which we will derive our analyses of popular model architectures, and also introduce our new neural model, the PossibleWorldNet. In Section 4, we describe the structure of experiments, from which we obtained the results presented and discussed in Section 5. We offer a brief survey of related work in Section 6, before making concluding remarks in Section 7.

## 2 DATASET CREATION

Formal logics provide a symbolic toolkit for encoding and examining patterns of reasoning. They are structural calculi aiming to codify the norms of correct thought. The meanings of such statements are invariant to what the particular propositions stand for: to understand the entailment $(p \wedge q) \vDash q$, we only need to understand the semantics of—or related syntactic rules governing—a finite set of logical connectives, while $p$ and $q$ are meaningless arbitrary symbols selected to stand for distinct propositions. In other words, the problem of determining whether an entailment holds is a purely structural sequence-based problem: to evaluate whether an entailment is true, only the meaning of—or inference rules governing—the connectives is relevant. Everything else only has meaning via its place in the structure specified by an expression. These qualities suggest that detecting logical entailment is an excellent task for measuring the ability of models to capture, understand, or exploit structure. We present in this paper a generic process for generating entailment datasets, explained in detail in Appendix A, for any given logical system. In the specific dataset—generated through this process—presented in this section, we will focus on propositional logic, which is decidable but requires a worst case of $O(2^n)$ operations (e.g. resolution steps, truth table rows), where $n$ is the number of unique propositional variables, to verify entailment.

Our dataset[*] $\mathcal{D}$ is composed of triples of the form $(A, B, A \vDash B)$, where $A$ and $B$ are formulas of propositional logic, and $A \vDash B$ is 1 if $A$ entails $B$, and 0 otherwise. For example, the data point $(p \wedge q, q, 1)$ is positive because $p \wedge q$ entails $q$, whereas $(q \vee r, r, 0)$ is negative because $q \vee r$ does not entail $r$. Entailment is primarily a semantic notion: $A$ entails $B$ if every model in which $A$ is true is also a model in which $B$ is true.

We impose various requirements on the dataset, to rule out superficial structural differences between $\mathcal{D}^+$ and $\mathcal{D}^-$ that can be easily exploited by "trivial" baselines[†]. We impose the following high level constraints on our data through the generative process, explained in detail in Appendix A: our classes must be balanced, and formulas in positive and negative examples must have the same distribution over length. Furthermore, we attempt to ensure that there are no recognisable differences in the distributions of lexical or syntactic features between the positive and negative examples. It would not be acceptable, for example, if a typical $B$ formula in a positive entailment $(A, B, 1)$ had more disjunctions than a $B'$ formula in a negative entailment $(A', B', 0)$.

If we simply sample formulas $A$ and $B$ and evaluate whether $A \vDash B$, there are significant differences between the distributions of formulas for the positive and negative examples, which models can learn to exploit without needing to understand the structure of the problem. To avoid these issues, we use a different approach, that satisfies the above requirements. We sample 4-tuples of formulas $(A_1, B_1, A_2, B_2)$ such that:

$$A_1 \vDash B_1 \qquad A_2 \vDash B_2 \qquad A_1 \nvDash B_2 \qquad A_2 \nvDash B_1$$

Here, each of the four formulas appears in one positive entailment and one negative entailment. This way, we minimise crude structural differences between the positive and negative examples. Here is a simple example (although the actual dataset has much longer formulas) of such a 4-tuple of datapoints:

$$p \vDash p \vee q \qquad \neg p \wedge \neg q \vDash \neg q \qquad p \nvDash \neg q \qquad \neg p \wedge \neg q \nvDash p \vee q$$

---

[*]We aim to release the dataset used for experiments, and the code used to generate it according to the constraints discussed in this paper, upon publication of the paper.

[†]E.g., bag-of-words baselines that cannot look at the structure of $A$ and $B$, or baselines that ignore $A$ and only look at $B$.

Table 1: Dataset Statistics

|  | Size | Mean # Vars | Mean # Ops | Mean Length | Mean $2^{\text{# Vars}}$ |
|---|---|---|---|---|---|
| **Train** | 100,000 | 4.5 | 5.3 | 11.3 | 52.2 |
| **Validate** | 5,000 | 5.1 | 6.8 | 13.0 | 75.7 |
| **Test (easy)** | 5,000 | 5.2 | 6.9 | 13.1 | 81.0 |
| **Test (hard)** | 5,000 | 5.8 | 17.4 | 31.5 | 184.4 |
| **Test (big)** | 5,000 | 8.0 | 20.9 | 38.7 | 3310.8 |
| **Test (massive)** | 2,230 | 18.4 | 49.4 | 88.8 | 848,570.0 |
| **Test (exam)** | 100 | 2.4 | 3.9 | 8.6 | 5.8 |

To generate these 4-tuples, we first generate pairs $(A, B)$ such that $A \vDash B$. (To test if $A \vDash B$, we test whether $A \wedge \neg B$ is satisfiable, using **minisat** (Sorensson & Een, 2005)). Then we search through the set of pairs, looking for pairs of pairs, $(A_1, B_1)$ and $(A_2, B_2)$, such that $A_1 \nvDash B_2$ and $A_2 \nvDash B_1$. We present, in Appendix A, the full details of this generative process, its constraints and guarantees, and how we used particular baselines to validate the data.

## 2.1 SPLITTING THE DATASET

We produced train, validation, and test (easy) by generating one large set of 4-tuples, and splitting them into groups of sizes 100000, 5000, and 5000. The difficulty of evaluating an entailment depends on the number of propositional variables and the number of operators in the two formulas. In training, validation, and test (easy), we sample the number of propositional variables uniformly between 1 and 10 (there are 26 propositional variables in total: $a$ to $z$). In test (hard), we sample uniformly between 5 and 10. Our formula sampling method takes a parameter specifying the desired number of operators in the formula. In training, validation, and test (easy), the number of operators in a formula is sampled uniformly between 1 and 10. In our hard test set, the number of operators in a formula is sampled uniformly between 15 and 20.

For the test (big) dataset, we sampled formulas using between 1 and 20 variables (uniformly), and between 10 and 30 operators (again, uniformly). For test (massive), we used a different generating mechanism. We first sampled pairs of formulas $A, B$ such that $A \models B$. These had between 20 and 26 variables, and between 20 and 30 operators each. Then we generated a $B^*$ by mutating $B$ and checking that $A \nvDash B^*$. See Table 1 for detailed statistics of the dataset sections, including the average difficulty (based on a complexity of $\mathcal{O}(2^{\text{# Vars}})$) of sequents in each fold.

The test (exam) dataset was assembled from 100 examples of logical entailment in the wild. We looked through various logic textbooks for classic examples of entailments. From these textbooks, we extracted true entailment triples $(A, B, 1)$ where $A \models B$. We added false triples $(A, B^*, 0)$, by mutating $B$ into $B^*$ and checking that $A \nvDash B^*$.

In order to test models' ability to generalise to new unseen formulas, we pruned out cases where formulas seen in validation and test were $\alpha$-equivalent (equivalent up to renaming of symbols) to formulas seen in training. So, for example, if it had seen $p \models (\neg q \wedge p)$ in training, we did not want $r \models (\neg s \wedge r)$ to appear in either the test or validation sets. To do this, we converted all formulas to de-Bruijn form (see Pierce (2002), Chapter 6), and filtered out formulas in validation and test whose de-Bruijn form was identical to one of those in training. This prevents the system from being able to simply memorise examples it has seen in training.

## 2.2 DATA AUGMENTATION THROUGH SYMBOLIC VOCABULARY PERMUTATION

As discussed above, the logical connectives $(\vee, \wedge, \dots)$ are the only elements of the language in each dataset that have consistent implicit semantics across expressions. In this sense, two entailments $p \wedge q \vDash q$ and $a \wedge b \vDash b$ should ideally be treated as identical by the model. To encourage models to capture this invariance, we add an optional data processing layer during training (not testing) whereby symbols are consistently replaced by other symbols of the same type within individual entailments before being input to the network according to the process described below. This is achieved by randomly sampling a permutation of $a, \dots, z$ (the propositional variables used) for

every training example, and applying this permutation to the left and right sequents. This process is analogous to augmenting image classification training with random reflections and crops.

## 3 Models

In this section, we first describe a couple of baseline models that verify the basic difficulty of the dataset, followed by a description of benchmark models which are commonly used (with some variation) in a variety of problems, and finally by a description of our new model, PossibleWorldNet.

### 3.1 Baselines

The classes in the dataset are balanced in training, validation, and both test sets, so a random baseline (and a constant, majority-class predicting baseline) will obtain an accuracy of 50% on the test sets.

We define two neural baselines which, we believe, should not be able to perform competitively on this task, but may do better than random. The first is a linear bag of words (Linear BoW) model which embeds each symbol to a vector, and averages them, to produce a representation of each side of the sequent. These representations are then passed through a linear layer:

$$P(A \vDash B) = \sigma \left( W \cdot \text{concat} \left( g(A), g(B) \right) + b \right) \quad \text{where} \quad g(X) = \frac{1}{|X|} \sum_{x \in X} \text{embed}(x)$$

The second is a similar architecture, where the final linear layer is replaced with a multi-layer perceptron (MLP BoW):

$$P(A \vDash B) = \sigma(\text{MLP}(\text{concat} \left( g(A), g(B) \right))) \quad \text{where} \quad g(X) = \frac{1}{|X|} \sum_{x \in X} \text{embed}(x)$$

In both of these cases, the baselines are expected to have limited performance since they can only capture entailment by modelling the contribution of symbols individually, rather than by modelling structure, since the summation in $g$ destroys all structural information (including word order). We use these results to provide an indication of the difficulty of the dataset.

### 3.2 Benchmarks

We present here a series of benchmark models, not only to serve the purpose of being grounds for comparison for new models tested against this dataset, but also to compare and contrast the performance of fairly ubiquitous model architectures on this purely syntactic problem.

We distinguish two categories of models: encoding models and relational models. Encoding models, with exceptions specified below, jointly learn an encoding function $f$ and an MLP, such that given a sequent $A \vDash B$, the model expresses

$$P(A \vDash B) = \sigma \left( \text{MLP}(\text{concat}(f(A), f(B))) \right).$$

In this sense $f$ produces a representation of each side of the sequent which contains all the information needed for the MLP to decide on entailment. In contrast, relational models will observe the pair of expressions and make a decision, perhaps by traversing both expressions, or by relating substructure of one expression to that of the other. These models express a more general formulation

$$P(A \vDash B) = \sigma \left( f(A, B) \right).$$

### 3.2.1 Encoder benchmarks

The first encoder benchmark implemented is a Deep Convolutional Network Encoder (ConvNet Encoders), akin to architectures described in the convolutional networks for text literature (Kalchbrenner et al., 2014; Zhang et al., 2015; Kim et al., 2016). Here, the encoder function $f$ is a stack of one dimensional convolutions over sequence symbols embedded by an embedding operation embedSeq, interleaved with max pooling layers every $k$ layers (which is a model hyperparameter), followed by $n$ (also a hyperparameter) fully connected layers:

$$f(X) = \text{MLP}(\text{Conv1D}_n(\ldots \text{maxPool}(\text{Conv1D}_k(\ldots \text{Conv1D}_1(\text{embedSeq}(X))\ldots))\ldots))$$

The second and third encoder benchmarks are an LSTM (Hochreiter & Schmidhuber, 1997) encoder network (LSTM Encoders), and its bidirectional LSTM variant (BiDirLSTM Encoders). For the LSTM encoder, we embed the sequence symbols, and run an LSTM RNN over them, ignoring the output until the final state:

$$f(X) = h_{\text{final}} \quad \text{where} \quad h_{\text{final}} = \text{LSTM}(\text{embedSeq}(X))$$

For the bidirectional variant, two separate LSTM RNNs LSTM$^{\leftarrow}$ and LSTM$^{\rightarrow}$ are run over the sequence in opposite directions. Their respective final states are concatenated to form a representation of the expression:

$$f(X) = \text{concat}(h_{\text{final}}^{\leftarrow}, h_{\text{final}}^{\rightarrow}) \quad \text{where} \quad h_{\text{final}}^{\leftarrow} = \text{LSTM}^{\leftarrow}(\text{embedSeq}(X))$$
$$\text{and} \quad h_{\text{final}}^{\rightarrow} = \text{LSTM}^{\rightarrow}(\text{embedSeq}(X))$$

The benchmarks described thus far do not explicitly condition on structure, even when it is known, as they are designed to traverse a sequence from left to right and model dependencies in the data implicitly. In contrast, we now consider encoder benchmarks which rely on the provision of the syntactic structure of the sequence they encode, and exploit it to determine the order of composition. This inductive bias, which may be incorrect in certain domains (e.g., where no syntax is defined) or difficult to achieve in domains such as natural language text (where syntactic structure is latent and ambiguous), is easy to achieve for logic (where the syntax is known). The experiments below will seek to demonstrate whether is a helpful inductive architectural bias.

The fourth and fifth encoding benchmarks are (tree) recursive neural networks (Tai et al., 2015; Le & Zuidema, 2015; Zhu et al., 2015; Allamanis et al., 2016), also known as TreeRNNs. These recursively encode the logical expression using the parse structure[‡], where leaf nodes of the tree (propositional variables) are embedded as learnable vectors, and each logical operator then combines one or more of these embedded values to produce a new embedding. For example, the expression $(\neg a) \vee b$ is parsed as the tree with leaves $a$ and $b$, a unary node $\neg$ (with input the embedding of $a$), and a binary node $\vee$ (with inputs the embeddings of $\neg a$ and $b$). Following Allamanis et al. (2016), the fourth encoding benchmark is a simple TreeRNN (TreeNet Encoders), where each operator 'op' concatenates its inputs to a vector $x$, and produces the output

$$p = \frac{h}{\|h\|_2} \quad \text{where} \quad h = W_1^{\text{op}} x + W_2^{\text{op}} \sigma(W_3^{\text{op}} x + b_3^{\text{op}}) + b_1^{\text{op}}.$$

The fifth and final encoding benchmark (TreeLSTM Encoders) is a variant of TreeRNNs which adapts LSTM cell updates. This helps capture long range dependencies and propagate gradient within the tree. Our implementation follows Tai et al. (2015), modified to have per-op parameters as per TreeRNNs (see, also, the work by Le & Zuidema (2015) and Zhu et al. (2015)).

### 3.2.2 RELATIONAL BENCHMARKS

In addition to these encoding benchmarks, we define a pair of relational benchmarks, following Rocktäschel et al. (2015). We will traverse the entire sequent with LSTM RNNs or bidirectional LSTM RNNs but concatenating the left hand side and right hand side sequences into a single sequence separated by a held-out symbol (effectively standing for $\vDash$ ). For the LSTM variant (LSTM Traversal), the model is:

$$P(A \vDash B) = \sigma(\text{MLP}(h_{\text{final}})) \quad \text{where} \quad h_{\text{final}} = \text{LSTM}(\text{embedSeq}(\text{join}(A, ``\vDash", B)))$$

For the bidirectional case (BiDirLSTM Traversal), the extension is

$$P(A \vDash B) = \sigma(\text{MLP}(h_{\text{final}}^{\leftrightarrow})) \quad \text{where} \quad h_{\text{final}}^{\leftrightarrow} = \text{concat}(h_{\text{final}}^{\leftarrow}, h_{\text{final}}^{\rightarrow})$$
$$\text{with} \quad h_{\text{final}}^{\leftarrow} = \text{LSTM}^{\leftarrow}(\text{embedSeq}(X))$$
$$\text{and} \quad h_{\text{final}}^{\rightarrow} = \text{LSTM}^{\rightarrow}(\text{embedSeq}(X))$$

---

[‡]Completely accurate parses of logical expressions are trivial to obtain, and these are provided to the model rather than learned.

### 3.2.3 THE TRANSFORMER BENCHMARK

We also benchmark the Transformer model, also known as *Attention Is All You Need* (Vaswani et al., 2017), which is a sequence-to-sequence model achieving state-of-the-art results in machine translation. As in the relational LSTM models, we concatenate and embed the sequents, but instead of separating the sequents by a held-out symbol, we add a learnable bias to the right sequent in this embedding. This augments the Transformer's method of adding timing signals to distinguishing symbols at different positions. We then decode a sequence of length 1 and apply a linear transformation to get the final entailment prediction logits.

## 3.3 THE POSSIBLEWORLDNET

In this section, we introduce our new model. Inspired by the semantic (model-theoretic) definition of entailment, we propose a variant on TreeNets that evaluates the pair of formulas in different "possible worlds".

Entailment is, first and foremost, a *semantic* notion. Given a set $\mathcal{W}$ of worlds,

$$A \models B \text{ iff for every world } w \in \mathcal{W}, \ sat(w, A) \text{ implies } sat(w, B)$$

Here $sat : World \times Formula \rightarrow Bool$ indicates whether a formula is satisfied in a particular world.

We shall first define a variant of $sat$ that produces *integers*, and then define another variant that operates on real values. First, define $sat_2 : World \times Formula \rightarrow \{0, 1\}$:

$$sat_2(w, A) = \mathbb{1}(sat(w, A))$$

Using $sat_2$, we can redefine entailment as:

$$A \models B \text{ iff } \forall w \in \mathcal{W} \ sat_2(w, A) \leq sat_2(w, B)$$

Assume we have a finite set of worlds $\mathcal{W} = \{w_1, ..., w_n\}$; then we can recast as:

$$P(A \models B) = \prod_{i=1}^{n} \mathbb{1}(sat_2(w_i, A) \leq sat_2(w_i, B)) \tag{1}$$

We are going to produce a relaxation of Proposition 1 by replacing $sat_2$ and $\leq$ with continuous functions. Assume we have a variant of $sat_2$ that produces vectors of real values:

$$sat_3 : World \times Formula \rightarrow \mathbb{R}^d$$

Assume we have a function $f : \mathbb{R}^d \times \mathbb{R}^d \rightarrow [0, 1]$ that generalises $\leq$ to vectors of real values. Now we can rewrite as:

$$P(A \models B) = \prod_{i=1}^{n} f(sat_3(w_i, A), sat_3(w_i, B)) \tag{2}$$

In our neural model, $f$ is implemented by a simple linear layer using learnable weights $W_f$ and $b_f$:

$$f(x, y) = \sigma(W_f \cdot \text{concat}(x, y) + b_f)$$

We use a set of random vectors to represent our worlds $\{w_1, ..., w_n\}$, where $w_i \in \mathbb{R}^k$ is a vector of length $k$ of values drawn uniformly randomly. We implement $sat_3$ using a simplified TreeNN (see Section 3.2) as described below. Since $sat_3$ depends on the particular world $w_i$ we are currently evaluating, we add an additional parameter to the TreeNN so that the embedder has access to the current world $w_i$. We add an additional weight matrix $W_4^{op}$ so that propositional variables can learn which aspect of the current world to focus on. If the formula is of the form $op(l, r)$, where $op$ is nullary (a propositional variable), unary (e.g., negation), or binary (e.g., conjunction), and $l$ and $r$ are the embeddings of the constituents of the expression, then

$$sat_3(w_i, op(l, r)) = \frac{h}{\|h\|_2} \quad \text{where} \quad h = \begin{cases} W_4^{op} w_i & \text{where } op \text{ is nullary (leaf)} \\ W_1^{op} x + b_1^{op} & \text{otherwise} \end{cases}$$

where $x = \text{concat}(l, r)$.

To evaluate whether $A \vDash B$, the PossibleWorldNet generates a set of imagined "worlds", and then evaluates $A$ and $B$ in each of those worlds. It is a form of "convolution over possible worlds". As we will see in Section 5, the quality of the model increases steadily as we increase the number of imagined worlds.

This architecture was inspired by semantic (model-theoretic) approaches to detecting entailment, but it does not encode any constraint on propositional logic *in particular* or formal logic in general. The procedure of evaluating sentences in multiple worlds, and combining those evaluations in one product, is just what "entailment" means; so we speculate that an architecture like this should, in principle, be equally applicable to other logics (e.g., intuitionistic logic, modal logics, first-order logic) and also to non-formal entailments in natural language sentences.

Abstracting away from the particular interpretation of these vectors as "worlds", this method generates $n$ copies of the model with shared weights, one for each vector $w_i$; each nullary operator learns a different projection on $w_i$. It makes predictions via a linear layer combining two representations, and then takes the product of the predictions as the overall prediction.

## 4 EXPERIMENTAL SETUP

For each encoder benchmark architecture, the parameters of the encoders for the left and right hand sides of the sequent are shared. The MLP which performs binary classification to detect entailment based on the expression representations produced by the encoders is model-specific (re-initialised for each model) and jointly trained. Symbol embedding matrices are also model-specific, shared across encoders, and jointly trained.

We implemented all architectures in TensorFlow (Abadi et al., 2016). We optimised all models with Adam (Kingma & Ba, 2014). We grid searched across learning rates in $[1e-5, 1e-4, 1e-3]$, mini-batch sizes in $[64, 128]$, and trained each model thrice with different random seeds. Per architecture, we grid-searched across specific hyperparameters as follows. We searched across 2 and 3 layer MLPs wherever an MLP existed in a benchmark, and across layer sizes in $[32, 64]$ for MLP hidden layers, embedding sizes, and RNN cell size (where applicable). Additionally for convolutional networks, we searched across a number of convolutional layers in $[4, 6, 8]$, across kernel size in $[5, 7, 9]$, across number of channels in $[32, 64]$, and across pooling interval in $[0, 5, 3, 1]$ (where 0 indicates no pooling). For the Transformer model, we searched across the number of encoder and decoder layers in the range $[6, 8, 10]$, dropout probability in the range $[0, 0.1, 0.5]$, and filter size in the range $[128, 256, 384]$. Finally, for all models, we ran them with and without the symbol permutation data augmentation technique described in Section 2.2.

As a result of the grid search, we selected the best model for each architecture against validation results, and record training, validation, and all test accuracies for the associated time step, which we present below.

## 5 RESULTS AND DISCUSSION

Experimental results are shown in Table 2. The test scores of the best performing overall model are indicated in bold. The test scores of the best performing model which does not have privileged access to the syntax or semantics of the logic (i.e. excluding TreeRNN-based models) are italicised. The best benchmark test results are underlined.

We observe that the baselines are doing better than random (8.2 points above for the easy test set, for the MLP BoW, and 2.6 above random for the hard test set). This indicates that there are some small number of exploitable regularities at the symbolic level in this dataset, but that they do not provide significant information.

The baseline results show that convolution networks and BiDirLSTMs encoders obtain relatively mediocre results compared to other models, as do LSTM and BiDirLSTM Traversal models. LSTM encoders is the best performing model which does not have privileged access to the syntax trees. Their success relative to BiDirLSTMs Encoders could be due to their reduced number of parameters guarding against overfitting, and rendering them easier to optimise, but it is plausible BiDirLSTMs Encoders would perform similarly with a more fine-grained grid search. Both tree-based models take

Table 2: Propositional Logic Model Accuracy.

|           | model              | valid | test (easy) | test (hard) | test (big) | test (massive) | test (exam) |
|-----------|--------------------|-------|-------------|-------------|------------|----------------|-------------|
| **baselines** | Linear BoW     | 52.6  | 51.4        | 50.0        | 49.7       | 50.0           | 52.0        |
|           | MLP BoW            | 57.8  | 57.1        | 51.0        | 55.8       | 49.9           | 56.0        |
| **benchmark models** | Transformer | 57.1 | 56.8     | 50.8        | 51.2       | 50.3           | 46.9        |
|           | ConvNet Encoders   | 59.3  | 59.7        | 52.6        | 54.9       | 50.4           | 54.0        |
|           | *LSTM Encoders*    | 68.3  | 68.3        | 58.1        | 61.1       | 52.7           | 70.0        |
|           | BiDirLSTM Encoders | 66.6  | 65.8        | 58.2        | 61.5       | 51.6           | 78.0        |
|           | TreeNet Encoders   | 72.7  | 72.2        | 69.7        | 67.9       | 56.6           | 85.0 |
|           | TreeLSTM Encoders | 79.1 | 77.8 | 74.2 | 74.2 | 59.3 | 75.0 |
|           | LSTM Traversal     | 62.5  | 61.8        | 56.2        | 57.3       | 50.6           | 61.0        |
|           | BiDirLSTM Traversal| 63.3  | 64.0        | 55.0        | 57.9       | 50.5           | 66.0        |
| **new model** | **PossibleWorldNet** | 98.7 | **98.6** | **96.7** | **93.9** | **73.4** | **96.0** |

the lead amongst the benchmarks, with the TreeLSTM being the best performing benchmark overall on both test sets. For most models except baselines, the symbol permutation data augmentation yielded 2–3 point increase in accuracy on weaker models (BiDirLSTM encoders and traversals, an convolutional networks) and between 7–15 point increases for the Tree-based models. This indicates that this data augmentation strategy is particularly well fitted for letting structure-aware models capture, at the representational level, the arbitrariness of symbols indicating unbound variables.

Overall, these results show clearly that models that exploit structure in problems where it is provided, unambiguous, and a central feature of the task, outperform models which must implicitly model the structure of sequences. LSTM-based encoders provide robust and competitive results, although bidirectionality is not necessarily always the obvious choice due to optimisation and overfitting problems. Perhaps counter-intuitively, given the results of Rocktäschel et al. (2015), traversal models do not outperform encoding models in this pair-of-sequences traversal problem, indicating that they may be better at capturing the sort of long-range dependencies need to recognise textual entailment better than they are at capturing structure in general.

We conclude, from these benchmark results, that tree structured networks may be a better choice for domains with unambiguous syntax, such as analysing formal languages or programs. For domains such as natural language understanding, both convolutional and recurrent network architectures have had some success, but our experiments indicate that this may be due to the fact that existing tasks favour models which capture representational or semantic regularities, and do not adequately test for structural or syntactic reasoning. In particular, the poor performance of convolutional nets on this task serves as a useful indicator that while they present the right inductive bias for capturing structure in images, where topological proximity usually indicates a joint semantic contribution (pixels close by are likely to contribute to the same "part" of an image, such as an edge or pattern), this inductive bias does not carry over to sequences particularly well (where dependencies may be significantly more sparse, structured, and distant)[§]. The results for the transformer benchmark indicate that while this architecture can capture sufficient structure for machine translation, allowing for the appropriate word order in the output, and accounting for disambiguation or relational information where it exists within sentences, it does not capture with sufficient precision the more hierarchical structure which exists in logical expressions.

The best performing model overall is the PossibleWorldNet, which achieves significantly higher results than the other models, with 99.3% accuracy on test (easy), and 97.3% accuracy on test (hard). This is as to be expected, as it has the strongest inductive bias. This inductive bias has two components. First, the model has knowledge of the syntactic structure of the expression, since it is a variant of a TreeNet. Second, inspired by the definition of semantic (model-theoretic) entailment in

---

[§]Related to this point, Kim et al. (2016) show that convolutional networks make for good character-level encoders, to produce word representations, which are in turn better exploited by RNNs. This is consistent with our interpretation of our results, since at the character level, topological distance is—like for images—a good indicator of semantic grouping (characters that are close are usually part of the same word or n-gram).

general, the model evaluates the pair of formulas in lots of different situations ("possible worlds") and combines the various results together in a product[¶].

The quality of the PossibleWorldNet depends directly on the number of "possible worlds" it considers (see Figure 1). As we increase the number of possible worlds, the validation error rate goes down steadily. Note that the data-efficiency also increases as we increase the number of worlds. This is because adding worlds to the model does not increase the number of model parameters—it just increases the number of different "possibilities" that are considered.

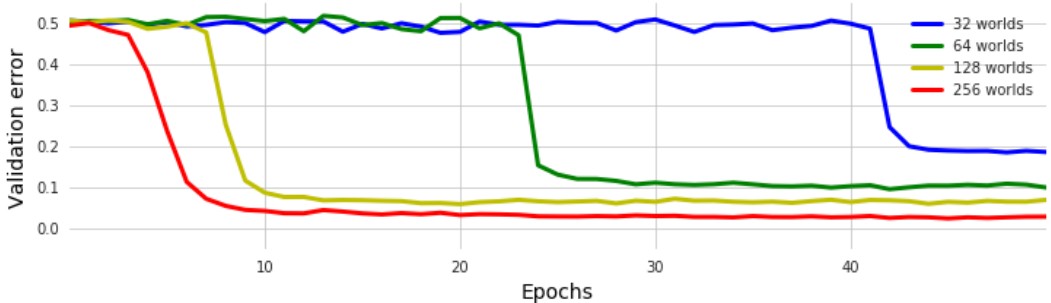

Figure 1: The quality of the PossibleWorldNet as we vary the number of possible worlds

In propositional logic, of course, if we are allowed to generate *every single* truth-value assignment, then it is trivial to detect entailment by checking each one. In our big test set, there are on average more than 3,000 possible truth-value assignments. In our massive test set, there are on average over 800,000 possible assignments. (See Table 1). The PossibleWorldNet considers at most 256 different worlds, which is only 7% of the expected total number of rows needed in the big test set, and only 0.03% of the expected number of rows needed for the massive test set.

To understand this result, we sample 32, 64, 128 and 256 truth table rows (variable truth-value assignments) for each pair of formulas in Test (hard), and reject entailment if a single evaluation for the formulas amongst these finds the left hand side to be true while the right hand side is false. This gives us an estimate of the accuracy of sampling a number of truth table rows equal to the number of possible worlds in our model. We estimate that these statistical methods have 75.9%, 86.5%, 93.4% and 97.2% chance of finding a countermodel, respectively. This seems to indicate that PossibleWorldNet is capable of exploiting repeated computation across projections of random noise in order to learn, solely based on the label likelihood objective, something akin to a model-based solution to entailment by treating the random-noise as variable valuations.

## 6 RELATED WORK

Zaremba et al. (2014) show how a neural architecture can be used to optimise matrix expressions. They generate all expressions up to a certain depth, group them into equivalence classes, and train a recursive neural network classifier to detect whether two expressions are in the same equivalence class. They use a recursive neural network (Socher et al., 2012) to guide the search for an optimised equivalent expression. There are two major differences between this work and ours. First, the classifier is predicting whether two matrix expressions (e.g. $A$ and $(A^T)^T$) compute the same values; this is an equivalence relation, while entailment is a *partial order*. Second, their dataset consists of matrix expressions containing at most one variable, while our formulas contain many variables.

Allamanis et al. (2016) use a recursive neural network to learn whether two expressions are equivalent. They tested on two datasets: propositional logic and polynomials. There are two main differences between their approach and ours. First, we consider *entailment* while they consider *equivalence*; equivalence is a symmetric relation, while entailment is not symmetric. Second, we consider entailment as a *relational* classification problem: given a pair of expressions $A$ and $B$, predict whether $A$ entails $B$. In their paper, by contrast, they generate a set of $k$ equivalence-classes of

---

[¶]See Formula 2 above. This general notion of entailment as truth-in-all-worlds is not dependent on any particular formal logic, and applies to entailment in both formal logics and natural languages.

formulas with the same truth-conditions, and ask the network to predict which of these $k$ classes a *single* formula falls into. Their task is more specific: their network is only able to classify a formula from a new equivalence class that has not been seen during training if it has additional auxiliary information about that class (e.g. exemplar members of the class).

Recognizing textual entailment (RTE) between *natural language sentences* is a central task in natural language processing. (See Dagan et al. (2006); for a recent dataset, see Bowman et al. (2015)). Some approaches (e.g., Wang & Jiang (2015) and Rocktäschel et al. (2015)) use LSTMs with attention, while others (e.g., Yin et al. (2015)) use a convolutional neural network with attention. Of course, recognizing entailment between natural language sentences is a very different task from recognizing entailment between logical formulas. Evaluating an entailment between natural language sentences requires understanding the meaning of the non-logical terms in the sentence. For example, the inference from *"An ice skating rink placed outdoors is full of people"* to *"A lot of people are in an ice skating park"* requires knowing the non-logical semantic information that an outdoors ice skating rink is also an ice skating park.

Current neural models do not always understand the structure of the sentences they are evaluating. In Bowman et al. (2015), all the neural models they considered wrongly claimed that *"A man wearing padded arm protection is being bitten by a German shepherd dog"* entails *"A man bit a dog"*. We believe that isolating the purely structural sub-problem will be useful because only networks that can reliably predict entailment in a purely formal setting, such as propositional (or first-order) logic, will be capable of getting these sorts of examples consistently correct.

## 7 CONCLUSION

In this paper, we have introduced a new process for generating datasets for the purpose of recognising logical entailment. This was used to compare benchmarks and a new model on a task which is primarily about understanding and exploiting structure. We have established two clear results on the basis of this task. First, and perhaps most intuitively, architectures which make explicit use of structure will perform significantly better than those which must implicitly capture it. Second, the best model is the one that has a strong architectural bias towards capturing the possible world semantics of entailment. In addition to these two points, experimental results also shed some light on the relative abilities of implicit structure models—namely LSTM and Convolution network-based architectures—to capture structure, showing that convolutional networks may not present the right inductive bias to capture and exploit the heterogeneous and deeply structured syntax in certain sequence-based problems, both for formal and natural languages.

This conclusion is to be expected: the most successful models are those with the most prior knowledge about the generic structure of the task at hand. But our dataset throws new light on this unsurprising thought, by providing a new data-point on which to evaluate neural models' ability to understand structural sequence problems. Logical entailment, unlike textual entailment, depends *only* on the meaning of the logical operators, and of the place particular arbitrarily-named variables hold within a structure. Here, we have a task in which a network's understanding of structure can be disentangled from its understanding of the meaning of words.

### ACKNOWLEDGMENTS

We thank our colleagues at DeepMind for their insightful comments during the preparation of this paper, and in particular Yujia Li, Chris Dyer, and Alex Graves.

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

# A  THE DATASET

## A.1  DATASET REQUIREMENTS

Our dataset $\mathcal{D}$ is composed of triples of the form $(A, B, A \vDash B)$, where $A \vDash B$ is 1 if $A$ entails[‖] $B$, and 0 otherwise. For example:

$$(p \wedge q, q, 1)$$
$$(q \vee r, r, 0)$$

We wanted to ensure that simple baseline models are unable to exploit simple statistical regularities to perform well in this task. We define a series of baseline models which, due to their structure or the information they have access to, should not be able to solve the entailment recognition problem described in this paper. We distinguish baselines for which we believe there is little chance of them detecting entailment, from those for which there categorically cannot be true modelling of entailment. The baselines which categorically cannot detect entailment are encoding models which only observe one side of the sequent:

$$P(A \vDash B) = \sigma\left(\text{MLP}(f(A))\right) \quad \text{or} \quad P(A \vDash B) = \sigma\left(\text{MLP}(f(B))\right)$$

where $f$ is a linear bag of words encoder, an MLP bag of words encoder, or a TreeNet.

Because the dataset contains a roughly balanced number of positive and negative examples, it follows that we should expect any model which only sees part of the sequent to perform in line with a random classifier. If they outperform a random baseline on test, there is a structural or symbolic regularity on one side (or both) which is sufficient to identify some subset of positive or negative examples. We use these baselines to verify the soundness of the generation process.

Let $\mathcal{D}^+$ and $\mathcal{D}^-$ be the positive and negative entailments:

$$\mathcal{D}^+ = \{(A, B) \mid (A, B, 1) \in \mathcal{D}\}$$
$$\mathcal{D}^- = \{(A, B) \mid (A, B, 0) \in \mathcal{D}\}$$

We impose various requirements on the dataset, to rule out superficial syntactic differences between $\mathcal{D}^+$ and $\mathcal{D}^-$ that can be easily exploited by the simple baselines described above. We require that our classes are balanced:

$$|\mathcal{D}^+| \quad = \quad |\mathcal{D}^-|$$

We do not want there to be any obvious difference in the length of formulas in $\mathcal{D}^+$ and $\mathcal{D}^-$:

$$\mathop{\mathbb{E}}_{(A,B)\sim\mathcal{D}^+} length(A) \quad = \quad \mathop{\mathbb{E}}_{(A,B)\sim\mathcal{D}^-} length(A)$$
$$\mathop{\mathbb{E}}_{(A,B)\sim\mathcal{D}^+} length(B) \quad = \quad \mathop{\mathbb{E}}_{(A,B)\sim\mathcal{D}^-} length(B)$$

We want there to be the same number of new free variables (variables appearing in B that do not appear in A) in both $\mathcal{D}^+$ and $\mathcal{D}^-$:

$$\mathop{\mathbb{E}}_{(A,B)\sim\mathcal{D}^+} |vars(B) - vars(A)| \quad = \quad \mathop{\mathbb{E}}_{(A,B)\sim\mathcal{D}^-} |vars(B) - vars(A)|$$

Let $num(A, op)$ be the number of occurrences of operator $op$ in formula $A$. So, for example, $num(\neg(p \wedge \neg q), \neg) = 2$. We impose the constraint that for each operator $op \in \{\neg, \wedge, \vee, \rightarrow\}$, that

$$\mathop{\mathbb{E}}_{(A,B)\sim\mathcal{D}^+} num(A, op) \quad = \quad \mathop{\mathbb{E}}_{(A,B)\sim\mathcal{D}^-} num(A, op)$$
$$\mathop{\mathbb{E}}_{(A,B)\sim\mathcal{D}^+} num(B, op) \quad = \quad \mathop{\mathbb{E}}_{(A,B)\sim\mathcal{D}^-} num(B, op)$$

---

[‖]Throughout, we focus on *classical* propositional logic, and do not consider e.g., intuitionistic entailment.

Furthermore, we require that the number of occurrences of an operator *at each level in the abstract syntax tree* is the same in $\mathcal{D}^+$ and $\mathcal{D}^-$. It would not be acceptable if, for example, a typical $B^+$ from $\mathcal{D}^+$ had more disjunctions at the top of the syntax tree than $B^-$ from $\mathcal{D}^-$. Let $num\_at(B, level, op)$ be the number of occurrences of operator $op$ at $level$ in the syntax tree for $B$. We also require that, for each $op$ and $level$:

$$\mathbb{E}_{(A,B)\sim\mathcal{D}^+} num\_at(A, level, op) = \mathbb{E}_{(A,B)\sim\mathcal{D}^-} num\_at(A, level, op)$$

$$\mathbb{E}_{(A,B)\sim\mathcal{D}^+} num\_at(B, level, op) = \mathbb{E}_{(A,B)\sim\mathcal{D}^-} num\_at(B, level, op)$$

## A.2  DATASET GENERATION

### A.2.1  A NAIVE APPROACH TO DATASET GENERATION

A simple way to generate an entailment dataset would be to alternate between first sampling formulas $A^+$ and $B^+$ such that $A^+ \vDash B^+$, and second sampling formulas $A^-$ and $B^-$ such that $A^- \nvDash B^-$. Since we are alternating between $\vDash$ and $\nvDash$, we are guaranteed to produce balanced classes. Unfortunately, this straightforward approach generates datasets that violate most of our requirements above. See Table 3 for the details.

In particular, the mean number of negations, conjunctions, and disjunctions at the top of the syntax tree ($num\_at(\cdot, 0, op)$) is markedly different. $A^+$ has significantly more conjunctions at the top of the syntax tree than $A^-$, while $B^+$ has significantly fewer than $B^-$. Conversely, $A^+$ has significantly fewer disjunctions at the top of the syntax tree than $A^-$, while $B^+$ has significantly more than $B^-$.

The mean number of satisfying truth-value assignments ($sat(\cdot)$) is also markedly different: $A^+$ is true in on average 3.7 truth-value assignments (i.e. it is a very specific formula which is only true under very particular circumstances), while $A^-$ is true in 10.3 truth-value assignments (i.e. it is true in a wider range of circumstances).

If we look at the mean number of variables appearing in $B$ that do not appear in $A$, there is also a striking difference between $\mathcal{D}^+$ and $\mathcal{D}^-$. The mean number of new variables in $vars(B^+) - vars(A^+)$ is 0.80 while the mean number of new variables in $vars(B^-) - vars(A^-)$ is 1.39 with a $\chi^2$ of 3308.1 and 8 degrees of freedom.

We can use these statistics to develop simple heuristic baselines that will be unreasonably effective on the dataset described above: we can estimate whether $A \vDash B$ by comparing the lengths of $A$ and $B$, or by looking at the number of variables in $B$ that do not appear in $A$, or by looking at the topmost connective in $A$ and $B$.

Table 3: Requirement violations in the naive approach, with $|\mathcal{D}| = 50,000$

| | $A^+$ | $A^-$ | $\chi^2$ | $\chi^2$ df | $B^+$ | $B^-$ | $\chi^2$ | $\chi^2$ df |
|---|---|---|---|---|---|---|---|---|
| $length(.)$ | 6.62 | 6.45 | 70.6 | 9 | 8.33 | 8.28 | 304.9 | 16 |
| $num(\cdot, \neg)$ | 1.47 | 1.33 | 309.4 | 8 | 1.77 | 1.91 | 139.0 | 9 |
| $num(\cdot, \wedge)$ | 1.52 | 1.33 | 308.6 | 8 | 1.70 | 1.94 | 134.0 | 11 |
| $num(\cdot, \vee)$ | 1.30 | 1.40 | 86.9 | 8 | 1.95 | 1.69 | 127.0 | 10 |
| $num\_at(\cdot, 0, \neg)$ | 0.31 | 0.22 | 532.4 | 1 | 0.18 | 0.30 | 350.9 | 1 |
| $num\_at(\cdot, 1, \neg)$ | 0.32 | 0.31 | 7.5 | 2 | 0.39 | 0.41 | 3.2 | 2 |
| $num\_at(\cdot, 2, \neg)$ | 0.31 | 0.31 | 8.8 | 4 | 0.56 | 0.54 | 5.3 | 4 |
| $num\_at(\cdot, 0, \wedge)$ | 0.35 | 0.2 | 1382.9 | 1 | 0.13 | 0.33 | 1076.4 | 1 |
| $num\_at(\cdot, 1, \wedge)$ | 0.32 | 0.31 | 36.5 | 2 | 0.39 | 0.40 | 6.5 | 2 |
| $num\_at(\cdot, 2, \wedge)$ | 0.31 | 0.32 | 3.2 | 4 | 0.56 | 0.53 | 16.5 | 4 |
| $num\_at(\cdot, 0, \vee)$ | 0.16 | 0.28 | 1070.3 | 1 | 0.34 | 0.16 | 752.4 | 1 |
| $num\_at(\cdot, 1, \vee)$ | 0.30 | 0.32 | 66.0 | 2 | 0.42 | 0.34 | 141.1 | 2 |
| $num\_at(\cdot, 2, \vee)$ | 0.32 | 0.31 | 12.9 | 4 | 0.57 | 0.52 | 39.7 | 4 |
| $\#sat(\cdot)$ | 3.7 | 10.3 | 11265 | 174 | 22.1 | 11.7 | 3702.8 | 241 |

### A.2.2 Our preferred approach to dataset generation

In order to satisfy our requirements above, we took a different approach to dataset generation. In order to ensure that there are no crude statistical measurements that can detect differences between $\mathcal{D}^+$ and $\mathcal{D}^-$, we change the generation procedure so that every formula appears in both $\mathcal{D}^+$ and $\mathcal{D}^-$. We sample 4-tuples of formulas $(A_1, B_1, A_2, B_2)$ such that:

$$
\begin{aligned}
A_1 &\vDash B_1 \\
A_2 &\vDash B_2 \\
A_1 &\nvDash B_2 \\
A_2 &\nvDash B_1
\end{aligned}
$$

Here, each of the four formulas appears in one positive entailment and one negative entailment[**].

Using this alternative approach, we are able to satisfy the requirements above. By construction, the mean length, number of operators at a certain level in the syntax tree, and the number of satisfying truth-value assignments is *exactly the same* for $\mathcal{D}^+$ and $\mathcal{D}^-$. See Table 4.

The only crude difference remaining is in the number of new variables. If we look at the number of variables appearing in $B$ that do not appear in $A$, there is a noticeable difference between $\mathcal{D}^+$ and $\mathcal{D}^-$. The mean number of new variables in $vars(B^+) - vars(A^+)$ is 1.25 while the mean number of new variables in $vars(B^-) - vars(A^-)$ is 1.60 with a $\chi^2$ of 922.1 and 8 degrees of freedom.

Table 4: Statistics for the preferred approach that generates 4-tuples, with $|\mathcal{D}| = 50,000$

| | $A^+$ | $A^-$ | $\chi^2$ | $\chi^2$ df | $B^+$ | $B^-$ | $\chi^2$ | $\chi^2$ df |
|---|---|---|---|---|---|---|---|---|
| $length(.)$ | 6.33 | 6.33 | 0.0 | 9 | 6.38 | 6.38 | 0.0 | 16 |
| $num(\cdot, \neg)$ | 1.42 | 1.42 | 0.0 | 9 | 1.26 | 1.26 | 0.0 | 8 |
| $num(\cdot, \wedge)$ | 1.63 | 1.63 | 0.0 | 7 | 1.16 | 1.16 | 0.0 | 7 |
| $num(\cdot, \vee)$ | 1.14 | 1.14 | 0.0 | 7 | 1.53 | 1.53 | 0.0 | 8 |
| $num\_at(\cdot, 0, \neg)$ | 0.33 | 0.33 | 0.0 | 1 | 0.16 | 0.16 | 0.0 | 1 |
| $num\_at(\cdot, 1, \neg)$ | 0.29 | 0.29 | 0.0 | 2 | 0.32 | 0.32 | 0.0 | 2 |
| $num\_at(\cdot, 2, \neg)$ | 0.30 | 0.30 | 0.0 | 3 | 0.31 | 0.31 | 0.0 | 4 |
| $num\_at(\cdot, 0, \wedge)$ | 0.49 | 0.49 | 0.0 | 1 | 0.1 | 0.1 | 0.0 | 1 |
| $num\_at(\cdot, 1, \wedge)$ | 0.34 | 0.34 | 0.0 | 2 | 0.31 | 0.31 | 0.0 | 2 |
| $num\_at(\cdot, 2, \wedge)$ | 0.30 | 0.30 | 0.0 | 4 | 0.30 | 0.30 | 0.0 | 4 |
| $num\_at(\cdot, 0, \vee)$ | 0.08 | 0.08 | 0.0 | 1 | 0.39 | 0.39 | 0.0 | 1 |
| $num\_at(\cdot, 1, \vee)$ | 0.27 | 0.27 | 0.0 | 2 | 0.35 | 0.35 | 0.0 | 2 |
| $num\_at(\cdot, 2, \vee)$ | 0.29 | 0.29 | 0.0 | 3 | 0.29 | 0.29 | 0.0 | 3 |
| $\#sat(\cdot)$ | 3.86 | 3.86 | 0.0 | 86 | 14.42 | 14.42 | 0.0 | 157 |

### A.3 Dataset Example

Our method generates 4-tuples such as the following:

$$
\begin{aligned}
p \vee p &\vDash (r \to c) \to ((r \to v) \vee p) \\
((g \vee p) \vee s) \to (g \to g) \wedge r &\vDash r \wedge (r \to r) \\
p \vee p &\nvDash r \wedge (r \to r) \\
((g \vee p) \vee s) \to (g \to g) \wedge r &\nvDash (r \to c) \to ((r \to v) \vee p)
\end{aligned}
$$

---

[**]One consequence of this method is that it rules out $A_1$ from being impossible (if it was impossible, we would not have $A_1 \nvDash B_2$) and $B_1$ from being a tautology (if it was a tautology, we would not have $A_2 \nvDash B_1$).

