# OpenReview forum: "Can Neural Networks Understand Logical Entailment?"
_ICLR.cc/2018/Conference — Accept (Poster)_

### Official Review · AnonReviewer3 · 2017-11-26
**Deep learning with entailment**

**Rating:** 7
**Confidence:** 3

**Review:**

SUMMARY

The paper is fairly broad in what it is trying to achieve, but the approach is well thought out. The purpose of the paper is to investigate the effectiveness of prior machine learning methods with predicting logical entailment and then provide a new model designed for the task. Explicitly, the paper asks the following questions: "Can neural networks understand logical formula well enough to detect entailment?", and "Which architectures are best at inferring, encoding, and relating features in a purely structural sequence-based problem?". The goals of the paper is to understand the learning bias of current architectures when they are tasked with learning logical entailment. The proposed network architecture, PossibleWorldNet, is then viewed as an improvement on an earlier architecture TreeNet.

POSITIVES

The structure of this paper was very well done. The paper attempts to do a lot, and succeeds on most fronts. The generated dataset used for testing logical entailment is given a constructive description which allows for future replication. The baseline benchmark networks are covered in depth and the reader is provided with a deep understanding on the limitations of some networks with regard to exploiting structure in data. The PossibleWorldNets is also given good coverage, and the equations provided show the means by which it operates.
• A clear methodological approach to the research. The paper covers how they created a dataset which can be used for logical entailment learning, and then explains clearly all the previous network models which will be used in testing as well as their proposed model.
• The background information regarding each model was exceptionally thorough. The paper went into great depth describing the pros and cons of earlier network models and why they may struggle with recognizing logical entailment.
• The section describing the creation of a dataset captures the basis for the research, learning logical entailment. They describe the creation of the data, as well as the means by which they increase the difficulty for learning.
• The paper provides an in depth description of their PossibleWorldNet model, and during experimentation we see clear evidence of the models capabilities.

NEGATIVES

One issue I had with the paper is regarding the creation of the logical entailment dataset. Not so much for how they explained the process of creating the dataset, that was very thorough, but the fact that this dataset was the only means to test the previous network models and their new proposed network model. I wonder if it would be better to find non-generated datasets which may contain data that have entailment relationships. It is questionable if their hand crafted network model is learned best on their hand crafted dataset.

The use of a singular dataset for learning logical entailment. The dataset was also created by the researchers for the express purpose of testing neural network capacity to learn logical entailment. I am hesitant to say their proposed network is an incredible achievement since PossibleWorldNet effectively beat out other methods on a dataset that they created expressly for it.

RELATED WORK

The paper has an extensive section dedicated to covering related work. I would say the research involved was very thorough and the researchers understood how their method was different as well as how it was improving on earlier approaches.

CONCLUSION

Given the thorough investigation into previous networks’ capabilities in logical entailment learning, I would accept this paper as a valid scientific contribution. The paper performs a thorough analysis on the limitations that previous networks face with regard to exploiting structure from data. The paper also covers results of the experiments by not only pointing out their proposed network’s success, but by analyzing why certain earlier network models were able to achieve competitive learning results. The structure of the PossibleWorldNet was also explained well, and during ex- perimentation demonstrated its ability to learn structure from data. The paper would have been improved through testing of multiple datasets, and not just on there self generated dataset, but the contribution of their research on their network and older networks is still justification enough for this paper.

---

> ### Author Response · Authors · 2017-12-22
> **Testing on "real world data"**
>
> Thank you for your supportive review and your kind comments. Based on questions you have raised with other reviewers, we have run further tests which we hope confirms your positive sentiment about the paper, and addressed any concerns you had about the testing regime used in the paper. We address here your specific, and very fair, criticism of our paper.
>
> “One issue I had with the paper is regarding the creation of the logical entailment dataset. Not so much for how they explained the process of creating the dataset, that was very thorough, but the fact that this dataset was the only means to test the previous network models and their new proposed network model. I wonder if it would be better to find non-generated datasets which may contain data that have entailment relationships. It is questionable if their hand crafted network model is learned best on their hand crafted dataset.”
>
> This is a good point. Since the initial submission, we have run a number of further experiments. In particular, we mined standard logic textbooks (e.g., Holbach’s “The Logic Manual”, Mendelson’s “Introduction to Mathematical Logic”) to find a set of entailment questions that were not produced from our synthetic generative process. We then held-out these entailments (and all entailments that were equivalent up to variable-renaming) from the training sets. The new test-set is called “Test (Exam)” in the revised Table 1 on page 3. We were gratified that our model achieved 96% on this “real-world” test-set. See the updated Table 2, reproduced below, including a variety of new larger test sets also described in this revision.
>
> +--------------------+-------+--------+--------+-------+-----------+--------+
> |                          |          | test   | test  | test | test       | test   |
> | model              | valid|easy  | hard | big  | mass.   | exam|
> +--------------------+-------+--------+--------+-------+-----------+--------+
> | Linear BoW    | 52.6  | 51.4 | 50.0   | 49.7 | 50.0     | 52.0   |
> +--------------------+-------+--------+--------+-------+-----------+--------+
> | MLP BoW        | 57.8  | 57.1 | 51.0   | 55.8  | 49.9    | 56.0   |
> +--------------------+-------+--------+--------+-------+-----------+--------+
> | ConvNet Enc. | 59.3  | 59.7  | 52.6  |54.9 | 50.4      | 54.0   |
> +--------------------+-------+--------+--------+-------+-----------+--------+
> | LSTM Enc.       | 68.3  | 68.3 | 58.1  | 61.1 | 52.7      | 70.0   |
> +--------------------+-------+--------+--------+-------+-----------+--------+
> | BiLSTM Enc.    | 66.6  | 65.8 | 58.2 | 61.5 | 51.6      | 78.0   |
> +--------------------+-------+--------+--------+-------+-----------+--------+
> | TreeNet Enc.  | 72.7  | 72.2  | 69.7 | 67.9 | 56.6      | 85.0   |
> +--------------------+-------+--------+--------+-------+-----------+--------+
> | TreeLSTM Enc| 79.1  | 77.8 | 74.2  | 74.2 | 59.3      | 75.0   |
> +--------------------+-------+--------+--------+-------+-----------+--------+
> | LSTM Trav.      | 62.5 | 61.8  | 56.2  | 57.3 | 50.6     | 61.0   |
> +--------------------+-------+--------+--------+-------+-----------+--------+
> | TreeLSTM Tr.  | 63.3  | 64.0 | 55.0  | 57.9 | 50.5     | 66.0   |
> +--------------------+-------+--------+--------+-------+-----------+--------+
> | PWN                | 98.7 | 98.6  | 96.7  | 93.9 | 73.4      | 96.0  |
> +--------------------+-------+--------+--------+-------+-----------+--------+

---

### Official Review · AnonReviewer2 · 2017-11-27
**A solved task?**

**Rating:** 4
**Confidence:** 4

**Review:**

This is a wonderful and a self-contained paper. In fact, it introduces a very important problem and it solves it.

The major point of the paper is demonstrating that it is possible to model logical entailment in neural networks. Hence, a corpus and a NN model are introduced. The corpus is used to demonstrate that the model, named PossibleWorld, is nearly perfect for the task. A comparative analysis is done with respect to state of the art recurrent NN. So far, so good.

Yet, what is the take home message? In my opinion, the message is that generic NN should not be used for specific formal tasks whereas specific neural networks that model the task are desirable. This seems to be a trivial claim, but, since the PossibleWorld nearly completely solves the task, it is worth to be investigated.

The point that the paper leaves unexplained is: what is in the PossibleWorld Network that captures what we need? The description of the network is in fact very criptic. No examples are given and a major effort is required to the reader. Can you provide examples and insights on why this is THE needed model?

Finally, the paper does not discuss a large body of research that has been done in the past by Plate. Plate has investigated how symbolic predicates can be described in distributed representations. This is strictly related to the problem this paper investigates. As discussed in "Symbolic, Distributed and Distributional Representations for Natural Language Processing in the Era of Deep Learning: a Survey", 2017, the link between symbolic and distributed representations has to be better investigated in order to propose innovative NN models. Your paper can be one of the first NN model that takes advantage of this strict link.

---

> ### Author Response · Authors · 2017-12-22
> **Revisions, extended test protocol, and some responses**
>
> Thank you for your kind words at the beginning of the review, and for your excellent questions and comments. We find the topics addressed in your questions, and your critical points are–we think–fair ones. We confess we were a little surprised by the low score given, considering the generally positive tone of the first half of the review, but this (along with the comments of other reviewers) has prompted us to rethink the evaluation of the models in order to address your specific points and hopefully assuage your concerns. We have made revisions to the manuscript to include further tests of our already-trained models, and we respond to parts of your review below.
>
> “The point that the paper leaves unexplained is: what is in the PossibleWorld Network that captures what we need? The description of the network is in fact very criptic. No examples are given and a major effort is required to the reader. “
>
> Limitations of space prevented us providing examples in the body of the text. Here is a simple example from propositional logic. To check whether p ∨ q entails p, we consider all possible truth-value assignments to the variables p and q. We get four assignments:
> p → ⊥, q → ⊥
> p → ⊥, q → ⊤
> p → ⊤, q → ⊥
> p → ⊤, q → ⊤
>
> Now  p ∨ q entails p if, for each of these assignments, if the assignment satisfies p ∨ q, then it also satisfies p. In this example, the entailment is false, since the second assignment (p → ⊥, q → ⊤) satisfies p ∨ q but  does not satisfy p.
>
> We will endeavour to add such examples to the appendix if the paper is accepted.
>
> “Can you provide examples and insights on why this is THE needed model?”
>
> Consider the standard model-theoretic definition of entailment: A entails B if, for every possible world w, if sat(w, A) then sat(w, B):
>
> A ⊧ B iff for every world w ∈ W, sat(w, A) implies sat(w, B)
>
> We replace possible worlds with random vectors, transform the universal quantification into a product, and provide a neural network that implements sat(w, A). The reason we believe this is *the* needed model is that it is a continuous relaxation of the standard model-theoretic definition of entailment.
>
> "In my opinion, the message is that generic NN should not be used for specific formal tasks whereas specific neural networks that model the task are desirable. This seems to be a trivial claim."
>
> At this high level of generality, the claim is, indeed, trivial. But our claim is more specific. We provide implementation details of a particular model that outperforms other models on this task. This model is also applicable outside the particular domain of logic.
>
> The PossibleWorldNet was inspired by the model-theoretic definition of entailment, in terms of truth in all possible worlds. But it is not a specific model that is only useful for this particular task. It is a general model based on the following simple idea: first, evaluate the same model multiple times using different vectors of random noise as inputs; second, combine the results from these multiple runs using a product. This general model is applicable outside the domain of logical entailment; it could be useful for building robust image classifiers, for example.
>
> Since the initial submission, we have run a number of experiments that are significantly more ambitious . See the updated Table 1 on page 3. In the “Big” and “Massive” test sets, the expected number of truth-table rows needed to exhaustively verify entailment is 3000 and 800,000. Our PossibleWorldNet continues to out-perform the other models on these harder test-sets, but it does not completely solve the task. In particular, in the massive test set, it achieves 73%. This score is significantly better than the other models, but it is not a complete solution.
>
> We hope that these explanations, which have been integrated into the revised manuscript, alongside the inclusion of further tests, without need to change the training protocol or model definitions, on significantly more complex logic problems and "real-world" exam data, will help convince you that this work merits publication.
>
> If you persist in your assessment, we will understand, but would be grateful if you could highlight what is lacking given this further empirical evidence provided in this revision, so that we may continue to improve the paper.

---

### Official Review · AnonReviewer1 · 2017-11-27
**The paper proposes a new model to use deep models for detecting logical entailment**

**Rating:** 7
**Confidence:** 3

**Review:**

Overall, the paper is well-written and the proposed model is quite intuitive. Specifically, the idea is to represent entailment as a product of continuous functions over possible worlds. Specifically, the idea is to generate possible worlds, and compute the functions that encode entailment in those worlds. The functions themselves are designed as tree neural networks to take advantage of logical structure. Several different encoding benchmarks of the entailment task are designed to compare against the performance of the proposed model, using a newly created dataset. The results seem very impressive with > 99% accuracy on tests sets.

One weakness with the paper was that it was only tested on 1 dataset. Also, should some form of cross-validation be applied to smooth out variance in the evaluation results. I am not sure if there are standard "shared" datasets for this task, which would make the results much stronger.
Also how about the tradeoff, i.e., does training time significantly increase when we "imagine" more worlds. Also, in general, a discussion on the efficiency of training the proposed model as compared to TreeNN would be helpful.
The size of the world vectors, I would believe is quite important, so maybe a more detailed analysis on how this was chosen is important to replicate the results.
This problem, I think, is quite related to model counting. There has been a lot of work on model counting. a discussion on how this relates to those lines of work would be interesting.


After revision

I think the authors have improved the experiments substantially.

---

> ### Author Response · Authors · 2017-12-22
> **Further tests, and clarifications for PWN**
>
> Thank you for your comments and fair criticisms. We have run substantial further evaluation of the previously trained models, which we hope will strengthen the case for this paper. We reply to some of the points you made in your review below, and hope you will find that the empirical evidence satisfactorily addressed the concerns you have raised.
>
> “One weakness with the paper was that it was only tested on 1 dataset. Also, should some form of cross-validation be applied to smooth out variance in the evaluation results. I am not sure if there are standard "shared" datasets for this task, which would make the results much stronger.
>
> This is a good point. To address this question, we have generated two other test sets. The first one, Test (big) has 1-20 variables and 10-30 operators per formula. The second, Test (massive) has 20-26 variables with 20-30 operators per formula. Finally, we collected a "real world" test set, Test (exam) from formulas found in textbook and exam questions, pruning sequents from the training set that were alpha-equivalent to sequents found in exam data. See the updated Table 1 for the new test-sets, and Table 2 for the updated results. In particular, there is still a gap between what is achieved by our best models and what is theoretically possible (> 25% accuracy gap) for the massive dataset, showing that further research on this topic is needed, and is hopefully enabled by this dataset.
>
> “Also how about the tradeoff, i.e., does training time significantly increase when we "imagine" more worlds. “
>
> Yes, the model takes longer to run (in terms of time) as we increase the number of worlds, since we need to evaluate the formulas in every world. But in terms of the number of training epochs, it does not take longer to run. One of the interesting things about the PossibleWorldNet is that the number of parameters (trainable variables) does not increase as we increase the number of worlds, nor does the model see more data. It just does more parallel computation per data point.
>
> “This problem, I think, is quite related to model counting. There has been a lot of work on model counting. a discussion on how this relates to those lines of work would be interesting.”
>
> Thanks, this is a good suggestion. We will certainly look into this for the final version.

---

### Public Comment · (anonymous) · 2017-11-03
**Some questions**

Dear authors,
Thank you for your nice work, I enjoyed reading your paper. I have several questions about your paper and I appreciate if you can answer them to make things more clear.

1. Variables in the formulas: in section 2.1 you mention that you have 26 variables in total. Does this mean that each formula has 26 variables? I suspect that this is not the case since you state in Table 1 that the average number of variables is 4.5 in the train set. If this assumption is correct, then can you point out how many of your formulas out of the 100,000 have 26 variables in them?

2. I noticed that your test sets do not have beyond 10 variables. Is there a particular reason why you don't test on the more complex formulas including more variables?

3. In section 5, the last paragraph, you compute the total number of possible truth-value assignments using 26 variables. If there are on average 4.5 variables in each formula, shouldn't the number of possible truth-values on average be $2^{4.5}$?

4. In section 3.3, you mention that $w_i \in \mathbb{R}^{k}$. Is $k$ equal to the number of variables in each formula? If so, does this mean that $w_i$ is a binary vector that indicates the possible values of each of the variables in the formula?

5. When creating Figure 1, which formulas were considered in training? If my understanding is correct in question 4, then you need formulas that have at least 8 variables to be able to generate 256 different worlds for each formula. This covers all possible configurations of the input formula for 8 variables. Is this a correct understanding?

6. As you have mentioned in the paper, TreeLSTMs reveal the best results among your benchmarks. But your possibleWorldNets use treeNets. Was there a particular reason you chose TreeNets? Did you also try out TreeLSTMs with your possibleWorldNets?

7. In section 5 you mention that the reason why BiDirLSTM is doing worse than LSTM Encoders is the fact that the BiDirLSTM is overfitting.  Was this the case for even less number of parameters for the BiDirLSTM? Is it also possible to conclude that the reason for BiDirLSTMs not being as effective, is the fact that it might not be useful to parse the formulas in reverse order for logical entailment? If I am not mistaken the reason why BiDirLSTMs do well compared to LSTMs, on e.g. NLP tasks is that words that appear later in the sentence might have a connection to words that appear earlier in the sentence. Is it correct to conclude that this is not the case for logical entailment?

---

> ### Author Response · Authors · 2017-11-16
> **Some Answers [1/4]**
>
> Thank you for these thoughtful questions, and for your patience in awaiting our response. This gives us the opportunity to clarify some of the things that should have been clearer in the paper. We copy your questions followed by our answers, for readability. These will be split across four comments due to OpenReview's comment character limits.
>
> 1. Variables in the formulas: in section 2.1 you mention that you have
> 26 variables in total. Does this mean that each formula has 26
> variables? I suspect that this is not the case since you state in
> Table 1 that the average number of variables is 4.5 in the train set.
> If this assumption is correct, then can you point out how many of your
> formulas out of the 100,000 have 26 variables in them?
>
> Yes, no formula has all 26 variables in it. We have a pool of 26 variables: {a, …, z} in total. A formula is generated by (i) first sampling between 1 and 10 propositional variables from {a, …, z}; call this temporary set of variables V and then (ii) generating a formula of the desired operator-complexity; each time the sampling procedure needs a variable, it chooses one of the variables in V. It is possible (in fact frequently the case) when generating a formula that the sampling procedure will not use all the variables in V.
>
> For example, we generate a set of 10 variables V = [b, c, d, j, m, p, q, s, v, x], and then generate a formula with 10 operators in it, using V. We get: ¬(((p ∨ m) ∧ (p ∨ d)) → ¬((j → c) ∧ ¬¬p))
> This only uses five of the ten variables in it.

---

> ### Author Response · Authors · 2017-11-16
> **Some Answers [2/4]**
>
> 2. I noticed that your test sets do not have beyond 10 variables. Is
> there a particular reason why you don't test on the more complex
> formulas including more variables?
>
> The only reason for choosing 1-10 variables in the paper was to limit the time it takes to generate the training/test data. Recall, that to generate a hard/unbiased dataset, we wanted to find four-tuples satisfying the four entailment conditions which ensures that the task cannot be solved with simply finding structural biases. As the number of variables increases, the chance of finding a four-tuple (A₁, B₁, A₂, B₂) satisfying the four conditions drops. Having said that, the 4-tuple constraint need not be imposed on test data, as it is only a useful constraint during training to prevent degenerate solutions.
>
> To address your question, we have generated two other test sets. The first one, Test (big) has 1-20 variables and 10-30 operators per formula. The second, Test (massive) has 20-26 variables with 20-30 operators per formula. Finally, we collected a "real world" test set, Test (exam) from formulas found in textbook and exam questions, pruning sequents from the training set that were alpha-equivalent to sequents found in exam data. This has the effect of removing some of the simplest examples from the training data, requiring us to re-run training and evaluation for all models. The statistics of these new datasets are found in the table below (which will hopefully stay somewhat formatted in the comment), which will replace Table 1 in future versions of the paper. We present and discuss model evaluation against the tests sets in our answer to question 3, below. As can be seen from the results, the proposed PWN model continues to be far superior to all other baselines on the more challenging datasets.
>
> +----------------+--------+--------+-------+--------+------------+
> |                     |           |Mean|Mean|Mean|Mean     |
> |                     | Size   |#Vars|#Ops|Len    |2^#Vars|
> +----------------+--------+--------+-------+--------+------------+
> | Train           |99,876| 4.5   | 5.3   | 11.3   | 52.2       |
> +----------------+--------+--------+-------+--------+------------+
> | Validate     | 5,000 | 5.1    | 6.8   | 13.0  | 75.7       |
> +----------------+--------+--------+-------+--------+------------+
> | Test (easy) | 5,000| 5.2    | 6.9   | 13.1   | 81.0       |
> +----------------+--------+--------+-------+--------+------------+
> | Test (hard)| 5,000 | 5.8    | 17.4 | 31.5  | 184.4     |
> +----------------+--------+--------+-------+--------+------------+
> | Test (big)   | 5,000  | 8.0  | 20.9 | 38.7  | 3310.8    |
> +----------------+--------+--------+-------+--------+------------+
> | Test (mass.)| 2,230| 18.4 | 49.4 | 88.8 |848,670.0|
> +----------------+--------+--------+-------+--------+------------+
> | Test (exam)| 100  | 2.4    | 3.9   | 8.6    | 5.8          |
> +----------------+--------+--------+-------+--------+------------+

---

> ### Author Response · Authors · 2017-11-16
> **Some Answers [3/4]**
>
> 3. In section 5, the last paragraph, you compute the total number of
> possible truth-value assignments using 26 variables. If there are on
> average 4.5 variables in each formula, shouldn't the number of
> possible truth-values on average be $2^{4.5}$?
>
> We acknowledge that the last paragraph in section 5 is misleading. This paragraph will be rewritten based on the new results below.
>
> We report, in Table 1, the mean number of variables for each section of dataset, as well as the mean number of rows needed to compute the truth tables for expressions from that section (2^{# Vars}). We note that the average number of rows is the average of 2^{# Vars}, rather than two to the power of the average number of variables. (The former is larger than the latter). At the end of the paragraph, we place an upper bound on the number of value assignments that the PossibleWorldNet (PWN) would need to iterate over (in the discrete case) to properly check each case.
>
> Your question serves to highlight that the average number of rows a truth table-based method would need to compute is, for our test sets, lower than the number of possible worlds the PWN views when making predictions. To this end, we have run-all models on larger test sets described in our answer to question 2, alongside an extra test set of human-answerable questions. We tabulate here the new results, which differ in places from the results presented in the paper as the training set has had simple examples removed from it.
>
> +--------------------+-------+--------+--------+-------+-----------+--------+
> |                          |          | test   | test  | test | test       | test   |
> | model              | valid|easy  | hard | big  | mass.   | exam|
> +--------------------+-------+--------+--------+-------+-----------+--------+
> | Linear BoW    | 52.6  | 51.4 | 50.0   | 49.7 | 50.0     | 52.0   |
> +--------------------+-------+--------+--------+-------+-----------+--------+
> | MLP BoW        | 57.8  | 57.1 | 51.0   | 55.8  | 49.9    | 56.0   |
> +--------------------+-------+--------+--------+-------+-----------+--------+
> | ConvNet Enc. | 59.3  | 59.7  | 52.6  |54.9 | 50.4      | 54.0   |
> +--------------------+-------+--------+--------+-------+-----------+--------+
> | LSTM Enc.       | 68.3  | 68.3 | 58.1  | 61.1 | 52.7      | 70.0   |
> +--------------------+-------+--------+--------+-------+-----------+--------+
> | BiLSTM Enc.    | 66.6  | 65.8 | 58.2 | 61.5 | 51.6      | 78.0   |
> +--------------------+-------+--------+--------+-------+-----------+--------+
> | TreeNet Enc.  | 72.7  | 72.2  | 69.7 | 67.9 | 56.6      | 85.0   |
> +--------------------+-------+--------+--------+-------+-----------+--------+
> | TreeLSTM Enc| 79.1  | 77.8 | 74.2  | 74.2 | 59.3      | 75.0   |
> +--------------------+-------+--------+--------+-------+-----------+--------+
> | LSTM Trav.      | 62.5 | 61.8  | 56.2  | 57.3 | 50.6     | 61.0   |
> +--------------------+-------+--------+--------+-------+-----------+--------+
> | TreeLSTM Tr.  | 63.3  | 64.0 | 55.0  | 57.9 | 50.5     | 66.0   |
> +--------------------+-------+--------+--------+-------+-----------+--------+
> | PWN                | 98.7 | 98.6  | 96.7  | 93.9 | 73.4      | 96.0  |
> +--------------------+-------+--------+--------+-------+-----------+--------+
>
> We observe that for the larger datasets, the tree-structured networks stay clearly ahead of the other benchmarks, and the PossibleWorldNet takes the top spot for all test sets, including the exams. For the test (huge) test set, there are nearly 8500,000 truth table rows to check on average, and yet the possible world net performs competitively with only 256 "worlds" per forward pass, showing that it is not replicating the brute-force verification method for detecting entailment within its activations. Even in the test (big) test set, with 3310 rows to check per sequent on average, a 256 world PWN nearly solves the problem.

---

> > ### Author Response · Authors · 2017-11-17
> > **Response Erratum**
> >
> > There is a typo in our response:
> > "For the test (huge) test set, there are nearly 8500,000 truth table rows"
> > This should read
> > "For the test (huge) test set, there are nearly 850,000 truth table rows"

---

> ### Author Response · Authors · 2017-11-16
> **Some Answers [4/4]**
>
> 4. In section 3.3, you mention that $w_i \in \mathbb{R}^{k}$. Is $k$
> equal to the number of variables in each formula? If so, does this
> mean that $w_i$ is a binary vector that indicates the possible values
> of each of the variables in the formula?
>
> $w_i$ is not a binary vector of truth-value assignments. It is a vector of reals, where each real value is uniformly sampled. Each world $w_i$ is just a vector of random noise.
>
> In an early experiment, we tried setting the $w_i$s to be vectors of Booleans, corresponding directly to truth-value assignments. We moved away from Boolean vectors because we wanted a neural model that was not tied specifically to propositional logic, that should be applicable to other logics (e.g. modal logics, first-order logic).
>
> The size of the world vectors, $k$, is a hyper-parameter. It is currently arbitrarily set to 26, but it doesn’t have to be. In future experimental runs, we would be curious to vary this hyper-parameter and plot performance as a function of its size, but have not had the time to do this yet.
>
> 5. When creating Figure 1, which formulas were considered in training?
> If my understanding is correct in question 4, then you need formulas
> that have at least 8 variables to be able to generate 256 different
> worlds for each formula. This covers all possible configurations of
> the input formula for 8 variables. Is this a correct understanding?
>
> The number of worlds is not determined by the number of variables in any particular formula. Rather, the number of worlds is a hyperparameter.
>
> 6. As you have mentioned in the paper, TreeLSTMs reveal the best
> results among your benchmarks. But your possibleWorldNets use
> treeNets. Was there a particular reason you chose TreeNets? Did you
> also try out TreeLSTMs with your PossibleWorldNets?
>
> We chose the TreeNet because it was the best performing benchmark we had. The central idea behind the PossibleWorldNet, implementing a continuous relaxation of model-checking, is applicable to any architecture. In future work, we plan to combine the PossibleWorldNet with other architectures, and compare the results.
>
> 7. In section 5 you mention that the reason why BiDirLSTM is doing
> worse than LSTM Encoders is the fact that the BiDirLSTM is
> overfitting.  Was this the case for even less number of parameters for
> the BiDirLSTM? Is it also possible to conclude that the reason for
> BiDirLSTMs not being as effective, is the fact that it might not be
> useful to parse the formulas in reverse order for logical entailment?
> If I am not mistaken the reason why BiDirLSTMs do well compared to
> LSTMs, on e.g. NLP tasks is that words that appear later in the
> sentence might have a connection to words that appear earlier in the
> sentence. Is it correct to conclude that this is not the case for
> logical entailment?
>
> We are uncertain how to interpret this particular result, as architecturally a bidirectional LSTM subsumes a unidirectional LSTM. We are not committed to this being an overfitting issue in particular, but clearly this architectural variant is more difficult to optimise in this setting. The same intuition as to why they work well as encoders in NLP tasks should apply here, and we hope further work will help to elucidate how to properly optimise bidirectional models on this task.

---

### Decision · Program_Chairs · 2018-01-29
**ICLR 2018 Conference Acceptance Decision**

**Decision:**

Accept (Poster)

**Comment:**

This paper studies the problem of modeling logical structure in a neural model.  It introduces a data set for probing various existing models and proposes a new model that addresses shortcomings in existing ones.  The reviewers point out that there is a bit of a tautology in introducing a new task and a new model that solves it.  The revised version addresses some of those concerns.  Overall, it is a thought-provoking and well-written study that will be interesting to discuss at ICLR.